# Mining Valuable Sub-Expressions for Symbolic Regression

## Abstract

Symbolic Regression (SR) aims to discover mathematical expressions from data, but classical methods are hampered by an immense search space. This inefficiency stems from their tendency to construct expressions atom-by-atom using basic operators and variables, overlooking the power of reusing meaningful sub-expressions. To address this challenge, we introduce Mining Sub-Expression Symbolic Regression (MSSR[1]), a novel framework that discovers and leverages valuable sub-expressions to efficiently search for the correct symbolic form. MSSR employs a cooperative multi-agent reinforcement learning framework, augmented with genetic programming, to intelligently sample sub-expressions from a dynamically evolving library, combining them into a mathematical expression. A pruning mechanism based on the coefficient of variation is utilized to remove redundant terms, promoting the discovery of the parsimonious expression. We conduct extensive experiments on the SRBench and fluid dynamics benchmarks. The results demonstrate that, compared to 24 baseline methods, MSSR recovers more ground-truth expressions and achieves a superior balance between predictive accuracy and model simplicity.

## 1 Introduction

Symbolic regression (SR) seeks to discover a mathematical expression $f^*$ from a vast search space $\Omega$ that best fits a given dataset $(X, y)$. Formally, the objective is to solve $f^* = \arg\min_{f \in \Omega} l(y, f(X))$, where $l$ is a loss function (Schmidt & Lipson, 2009; Korns, 2013). This task is fundamentally challenging; the SR problem is NP-hard (Virgolin & Pissis, 2022), and its discrete, non-differentiable, and extremely large search space makes finding the correct expression a formidable undertaking.

Current approaches to SR can be broadly categorized into two dominant paradigms: evolutionary algorithms and machine learning (ML) or deep learning (DL) methods. Evolutionary methods, a classic approach to SR, explore the search space using stochastic operations such as recombination and mutation to iteratively refine a population of candidate expressions (Ferreira, 2001; Koza, 1994; Schmidt & Lipson, 2009; Burlacu et al., 2020; Virgolin et al., 2017; Miller, 2020). In parallel, ML/DL techniques have emerged as a powerful alternative (Biggio et al., 2021; Kim et al., 2020; Kusner et al., 2017; Udrescu & Tegmark, 2020; Li et al., 2024). These methods typically follow one of two strategies. The first uses a neural network (NN) to parameterize a probability distribution over basic operators (e.g., $+, -, \times, x_1, \cdots$), from which an expression is sequentially sampled (Petersen et al., 2019; Landajuela et al., 2022). The second strategy embeds symbolic properties into the NN architecture itself, using basic operations as activation functions. After training on the dataset, the network's structure and non-zero weights can be directly interpreted as the final mathematical expression (Martius & Lampert, 2016; Sahoo et al., 2018).

A key limitation of the aforementioned methods is their reliance on basic operators to gradually construct mathematical expressions, often overlooking the significance of reusing meaningful sub-expressions. This oversight leads to a search space with a complexity of approximately $\Omega \approx |O|^n$, where $|O|$ denotes the number of operators and $n$ is the maximum length of the mathematical expression. However, if valuable sub-expressions are identified and used as building blocks, the search space can be reduced to approximately $|P|^{\frac{n}{k}}$, where $|P|$ is the number

---

[1]Code at supplementary material.

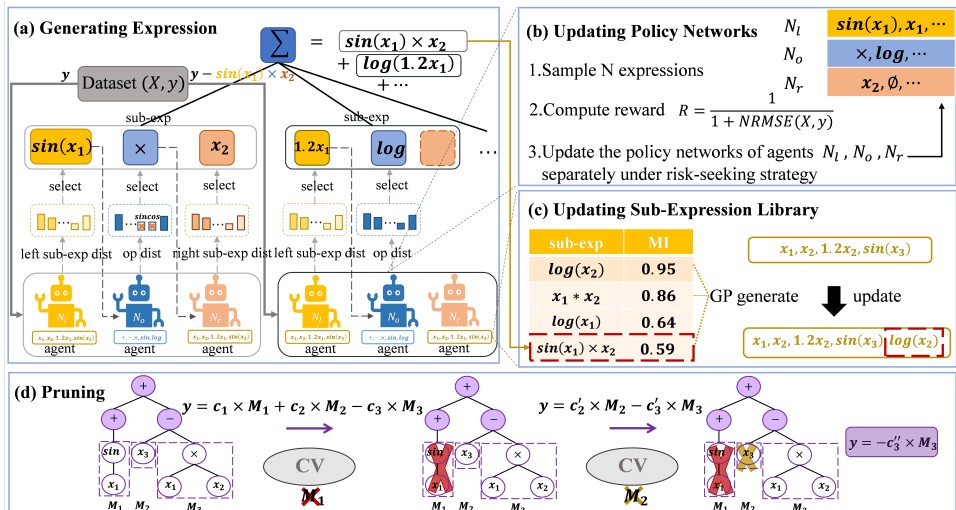

Figure 1: MSSR framework. (a) MSSR generates expressions by sampling the left sub-expressions, operator, and right sub-expressions through three agents $N_l$, $N_o$, and $N_r$. (b) MSSR then computes the reward of the sampling expressions and updates the policy networks of the three agents. (c) MSSR updates the sub-expression library using genetic programming (GP) with mutual information (MI). (d) MSSR prunes the redundant sub-expressions via the variation coefficient (CV).

of sub-expressions and $k$ is their average length. If the library size $|P|$ is constrained such that $|P| \ll |O|^k$, the resulting search space $|P|^{\frac{n}{k}}$ becomes significantly smaller than $|O|^n$. For example, consider the expression $x_1 + \cos(x_2 \times x_3)$ of length 6. Using a library of 9 operators and variables, $\{+, -, \times, \div, sin, cos, x_1, x_2, x_3\}$, the search space is approximately $\Omega \approx 9^6$. By incorporating $x_2 \times x_3$ into the library, its size becomes 10, and the expression length is reduced to 4, making the search space $\Omega \approx 10^4$. Thus, integrating valuable sub-expressions into the library can greatly reduce the search space $\Omega$.

To capitalize on this insight, we propose a novel approach called Mining Sub-Expression Symbolic Regression (MSSR), as illustrated in Fig. 1. First, MSSR represents the search space $\Omega$ as a sum of terms $\sum_n S_i$, where each term $S_i$ is a sub-expression composed as $S = \langle S_l, O, S_r \rangle$. MSSR leverages a cooperative multi-agent reinforcement learning framework to gradually generate the full expression $\sum_n S_i$. To construct each sub-expression $S_i$, three agents with different policy networks are responsible for sampling the left and right components, $S_l$ and $S_r$, from a dynamic sub-expression library $\mathcal{L}$, and an operator $O$ from a set $\{+, -, sin, \cdots\}$. For example, as shown in Fig. 1 (a), MSSR first samples a sub-expression $S_1 = sin(x_1) \times x_2$, where $S_l$ is '$sin(x_1)$', $O$ is '$\times$', and $S_r$ is '$x_2$'. The complete expression $\sum_n S_i$ is then constructed sequentially, with each term $S_i$ chosen greedily according to $\arg\min_{S_i \in \mathcal{L}} l(y, \sum_{k=1}^{i-1} S_k + S_i)$. For example, in Fig. 1 (a), the subsequent sub-expression $S_2 = log(1.2 \times x_1)$ is sampled based on the residual $y - sin(x_1) \times x_2$.

To improve the sampling of valuable sub-expressions, MSSR updates each agent's policy network using a risk-seeking policy gradient. In parallel, MSSR uses genetic programming (GP) to discover additional sub-expressions, which are evaluated based on their mutual information (MI) with the target. The library $\mathcal{L}$ is then updated with the sub-expressions possessing the highest MI, increasing its potential to contain valuable components, as shown in Figure 5. Finally, the complete expression $\sum_n S_i$ is pruned by evaluating each term $S_i$ with the coefficient of variation (CV) (Abdi, 2010) and removing those with lower contributions to simplify the final result.

The main contributions of this paper are as follows. 1) We propose MSSR, a framework for symbolic regression that efficiently utilizes sub-expressions to generate expressions that fit a given dataset. 2) We construct a cooperative multi-agent reinforcement learning model with three agents, which are respectively tasked with sampling the left sub-expressions, operators, and the right sub-expressions to form a complete sub-expression. 3) We demonstrate that MSSR exhibits strong competitiveness on the Feynman symbolic regression benchmarks (Udrescu & Tegmark, 2020), the Strogatz bench-

marks (La Cava et al., 2016), and the Penn machine learning benchmarks (Olson et al., 2017). It is one of the top methods for balancing accuracy and simplicity. Additionally, MSSR accurately discovers the core structure of general partial differential equations from multiple fluid datasets.

## 2 RELATED WORK

### 2.1 SYMBOLIC REGRESSION

Symbolic regression (SR) refers to discovering a mathematical expression fitted by the given dataset from the vast search space (Schmidt & Lipson, 2009; Korns, 2013). Genetic Programming (GP) (Koza, 1994; McKay et al., 2010; Ferreira, 2001; Miller & Turner, 2015; Arnaldo et al., 2014) is a key method for symbolic regression, encoding expressions and evolving them through operators like crossover, mutation, and selection to achieve better results. However, classic GP often treats each expression holistically, lacking a mechanism to explicitly identify and reuse valuable sub-expressions or structural motifs. This can lead to an inefficient, unguided search through the vast solution space. More recent ML/DL methods (Biggio et al., 2021; Kim et al., 2020; Sahoo et al., 2018; Kingma & Welling, 2013; Zhang & Zhou, 2021) use neural networks to generate expressions. One approach, exemplified by DSR (Petersen et al., 2019), treats expressions as pre-order sequences and uses a recurrent neural network (RNN) to generate them token-by-token. Another strategy, used by EQL (Martius & Lampert, 2016; Sahoo et al., 2018), replaces standard activation functions with basic operators (e.g., $+, \times, \sin$) to directly represent mathematical expressions within the network's architecture. Despite their novelty, these methods still construct solutions from atomic components and can suffer from unstable training or fail to find the ground truth in the large search space. In contrast, MSSR addresses this shared limitation by reducing the search space through the reuse of effective sub-expressions, enhancing the likelihood of finding the correct expression. ADF is used to achieve code reuse and modular development, similar to the concept of reusable sub-expressions (Ferreira, 2006). Some ADF-based methods may lead to multi-layer nesting (Ferreira, 2006; Zhong et al., 2015), resulting in structural redundancy. However, MSSR algorithm avoids this defect by focusing on mining increasingly meaningful sub-expressions with high mutual information.

### 2.2 ENHANCING REINFORCEMENT LEARNING VIA GENETIC PROGRAMMING (GP)

Reinforcement learning (RL) (Kaelbling et al., 1996) studies how an agent optimizes its decisions to maximize rewards. Cooperative multi-agent reinforcement learning (MARL) allows multiple agents to optimize a shared goal, extending single-agent reinforcement learning (Oroojlooy & Hajinezhad, 2023). In MSSR, each agent is responsible for decisions in different parts of the sub-expression and uses the same reward for optimization.

The synergy between the guided search of RL and the stochastic exploration of Genetic Programming (GP) has recently inspired a growing body of hybrid approaches. These methods generally fall into two categories: those that use RL to guide the evolutionary process of GP, and those that incorporate GP as a powerful exploration mechanism within an RL framework, which is the focus of our work. The inherent stochasticity of GP's crossover and mutation operators allows for diverse structural and semantic changes to expressions, making it well-suited for integration. For example, DSR-GP (Mundhenk et al., 2021) and uDSR (Landajuela et al., 2022) exemplify the first category by using a neural policy to generate a high-quality initial population for a subsequent GP search. Similarly, RSRM (Xu et al., 2024), which is based on MCTS and double Q-learning, also leverages evolutionary principles, applying GP to optimize expression trees generated by a separate process. Our work, MSSR, aligns with this second paradigm of using GP to enhance a primary learning algorithm. We employ GP as a complementary tool to introduce structural diversity into our library of sub-expressions, thereby augmenting the RL agent's ability to discover novel and effective symbolic building blocks.

### 2.3 MUTUAL INFORMATION AND COEFFICIENT OF VARIATION

Mutual information (MI) (Kraskov et al., 2004) and the coefficient of variation (CV) (Abdi, 2010) are core metrics used in the MSSR framework to measure correlation and stability. MI is used to measure the dependency between two random variables and remains unchanged under invertible

transformations (Zojaji & Ebadzadeh, 2016). MSSR uses MI to measure the dependency between sub-expressions and the target, and updates the sub-expression library with those showing higher MI. And, CV is the ratio of standard deviation to the mean Abdi (2010), which can be used to measure stability. MSSR leverages CV to identify and remove low-value sub-expressions, thus reducing redundancy.

## 3 MINING SUB-EXPRESSION SYMBOLIC REGRESSION

### 3.1 MODELING EXPRESSIONS

In MSSR, the search space $\Omega$ is defined as the sum of $n$ sub-expressions:

$$\Omega = \sum_{t=1}^{n} S_t = \sum_{t=1}^{n} \langle S_{l_t},\ O_t,\ S_{r_t} \rangle \tag{1}$$

Here, each term $S_t$ is the $t$-th sub-expression, consisting of a left sub-expression $S_{l_t}$, an operator $O_t$, and a right sub-expression $S_{r_t}$. For example, as illustrated in Fig. 1, the first sub-expression $S_1$ is represented as $\langle sin(x_1),\ \times,\ x_2 \rangle$, where '$sin(x_1)$' is the left sub-expression $S_{l_1}$, '$\times$' is the operator $O_1$, and '$x_2$' is the right sub-expression $S_{r_1}$.

The symbolic regression (SR) problem is to search this space $\Omega$ to find an expression that best fits the given dataset, i.e., $f^* = \arg\max_{f \in \Omega} R(y, f(X))$, where $R$ is the reward function. MSSR formalizes the SR problem as a cooperative **Multi-Agent Reinforcement Learning** (MARL) system $\langle \mathcal{N}, \mathcal{S}, \mathcal{A}, \mathcal{R} \rangle$. In this system, $\mathcal{N} = \{\mathcal{N}_l, \mathcal{N}_o, \mathcal{N}_r\}$ comprises three specialized agents that collaboratively generate a sub-expression $\langle S_l,\ O,\ S_r \rangle$; $\mathcal{S}$ is the state space; $\mathcal{A} = \{\mathcal{A}_l, \mathcal{A}_o, \mathcal{A}_r\}$ is the set of action spaces for all agents, where $\mathcal{A}_o = \{+, \times, sin, \cdots\}$, and the action spaces $\mathcal{A}_l$ and $\mathcal{A}_r$ are defined by the library $\mathcal{L} = \{x_1, x_2, 1.2 \times x_2, sin(x_3), \cdots\}$; and $\mathcal{R}$ is the reward function.

The cooperative MARL framework is governed by a joint policy $\boldsymbol{\pi}$, defined as the collection of individual policies: $\boldsymbol{\pi} = \{\pi_i(a_i|s_i; \theta_i) \mid i \in \{l, o, r\}\}$. $\pi_l$ selects the left sub-expression $S_l$, $\pi_o$ selects the operator $O$, and $\pi_r$ selects the right sub-expression $S_r$, which together compose a sub-expression $\langle S_l,\ O,\ S_r \rangle$. The policy network of each agent $\mathcal{N}_i$, parameterized by $\theta_i$, outputs a probability distribution over the action space $\mathcal{A}_i$ based on the state $s_i$. The general learning objective is to optimize $\boldsymbol{\pi}$ to maximize the expected reward:

$$\boldsymbol{\pi}^* = \arg\max_{\boldsymbol{\pi}} \mathbb{E}_{\boldsymbol{\pi}} [R_f] \tag{2}$$

where the reward $R_f$ is calculated from the fitness of the complete expression $f$ generated by the joint policy $\boldsymbol{\pi}$. However, in SR, the goal is not just to increase the average reward, but to find the best-performing expressions. Thus, the objective is modified to optimize for the top $\epsilon$-fraction of outcomes (Petersen et al., 2019):

$$\boldsymbol{\pi}^* = \arg\max_{\boldsymbol{\pi}} \mathbb{E}_{\boldsymbol{\pi}} [R_f \mid R_f \geq R_\epsilon]. \tag{3}$$

### 3.2 GENERATING EXPRESSIONS

Under the cooperative MARL framework, the mathematical expression $\sum_{t=1}^{n} S_t$ is generated step by step. At each time step $t$, the agents $\mathcal{N}_l, \mathcal{N}_o, \mathcal{N}_r$ select actions in turn to form the sub-expression $\langle S_{l_t},\ O_t,\ S_{r_t} \rangle$ based on the policy $\boldsymbol{\pi}$. Notably, the action selection for each agent $\mathcal{N}_i$ is based on a distinct local state, defined as $s_{l,t} = f_t$, $s_{o,t} = (f_t, S_{l_t})$, $s_{r,t} = (f_t, S_{l_t}, O_t)$. Here, $f_t$ encodes features of the current target value $y_t$ (mean, standard deviation, median, Pearson and Spearman correlation (Van Dongen & Enright, 2012)). When a unary operator is selected for $S_{l_t}$ (such as $sin, cos, \cdots$), the term $S_{r_t}$ is masked to maintain syntactic correctness.

After obtaining the partial expression $\sum_{k=1}^{t} S_k$, MSSR first optimizes any constants $C$ within it using L-BFGS, with the optimizer warm-started from the constants obtained at the previous time step. MSSR then evaluates the expression's output $y'_t$ and updates the target for the next step as $y_{t+1} \leftarrow y - y'_t$, which is used to construct the state $s_{t+1}$. This process can be seen as greedily fitting the residual at each time step. At the first time step, the initial target $y_1$ corresponds to $y$ from the given dataset $\{X, y\}$.

### 3.3 Updating Policy Networks

To optimize the joint policy $\boldsymbol{\pi}$ for generating more accurate mathematical expressions, MSSR adopts a Monte Carlo gradient estimation technique. Specifically, MSSR samples $N$ expressions and computes the reward $R$ for each. Based on the given dataset $X, y$, the reward $R$ is defined as follows (Petersen et al., 2019):

$$R = \frac{1}{1 + NRMSE(X, y)} \tag{4}$$

where $NRMSE(X, y) = \frac{1}{\sigma_y}\sqrt{\frac{1}{n}\sum_{i=1}^{n}(y_i - f(X_i))^2}$, $\sigma_y$ is the standard deviation of $y$, and $n$ is the dataset size. Consistent with the Independent Learners (IL) paradigm in cooperative MARL (Oroojlooy & Hajinezhad, 2023), the three agents ($\mathcal{N}_l, \mathcal{N}_o, \mathcal{N}_r$) share the same global reward $R$ and independently optimize their respective policy networks.

Let $p_i(f_i^{(k)})$ denote the probability of the $i$-th agent's policy network, parameterized by $\theta_i$, generating its trajectory component $f_i^{(k)}$ for the $k$-th expression. The probability $p_i(f_i^{(k)})$ decomposes into the product of selection probabilities over $n$ time steps as follows:

$$p_i(f_i^{(k)} \mid \theta_i) = \prod_{t=1}^{n} \pi_{i_t}(a_{i_t}^{(k)} \mid s_{i_t}^{(k)}; \theta_i). \tag{5}$$

Inspired by the risk-seeking policy gradient (Petersen et al., 2019), MSSR updates each policy network using the agent gradient derived in Proposition 1.

**Proposition 1** (Agent Gradient). *The gradient of the cooperative multi-agent objective is given by*

$$\nabla_{\theta_i} J_{risk}(\theta) \approx \frac{1}{\varepsilon N} \sum_{k=1}^{N} \left[ R_{f^{(k)}} - \tilde{R}_\varepsilon \right] \cdot \mathbf{1}_{R_{f^{(k)}} \geq \tilde{R}_\varepsilon} \nabla_{\theta_i} \log p_i(f_i^{(k)} \mid \theta_i) \tag{6}$$

Here, $\varepsilon$ represents the risk-seeking quantile threshold. The proof is provided in Appendix A.5. Additionally, MSSR adds an entropy gradient to encourage exploration.

### 3.4 Updating Sub-Expression Library

MSSR dynamically updates the sub-expression library $\mathcal{L}$ every $K$ generations to enhance the expressive power of the search. Specifically, at the end of the $K$ generations, MSSR uses the sampled expressions as an initial population for genetic programming to discover additional valuable sub-expressions, which are then evaluated by their mutual information (MI).

To update the library, MSSR applies two complementary selection strategies to balance diversity and quality. First, to ensure structural diversity, it selects the sub-expression with the highest MI for each operator type. Second, to maintain strong semantic relevance, it selects the sub-expressions with the highest overall MI scores. The number of sub-expressions added in each update is proportional to the number of operators in $\mathcal{L}$. For instance, the sub-expression '$log(x_2)$' might be added because it achieves a higher MI than other candidates, as shown in Fig. 1(c).

While an updated library $\mathcal{L}$ allows for more expressive solutions, it can also increase their complexity. To manage this trade-off, MSSR progressively reduces the number of terms, $n$, used to build the final expression as the library grows more powerful. The total number of library updates is limited to $n-1$, with $n$ decreasing by one after each update. Furthermore, the update interval $K$ is gradually increased over time to improve search efficiency.

### 3.5 Pruning Expressions

To simplify the final mathematical expression, MSSR uses the coefficient of variation (CV) (Abdi, 2010) to identify and prune insignificant terms. First, the sampled expression is represented as a binary expression tree and split into $N_m$ additive terms based on the '$+$' and '$-$' operators.

The importance of each term $S_i$ is assessed by measuring the stability of the remaining model in its absence. To calculate the CV associated with term $S_i$, it is temporarily removed from the expression. The given dataset $D$ is then randomly divided into $N_d$ subdatasets, $\{D_k | k = 1, 2, \cdots, N_d\}$.

For each subdataset $D_k$, a linear model is fit using the remaining terms. Let $c^i_{j,k}$ be the resulting coefficient of the $j$-th term when fitting on subdataset $D_k$ after term $S_i$ has been removed. The CV for term $S_i$ is defined as the sum of the CVs of the remaining coefficients:

$$CV_i = \sum_{j=1, j \neq i}^{N_m} \frac{\sigma(\xi^i_j)}{\mu(\xi^i_j)} \tag{7}$$

where $\xi^i_j$ is the set of coefficients $\left\{ c^i_{j,k} | k = 1, 2, \cdots, N_d \right\}$, and $\sigma$ and $\mu$ denote its standard deviation and mean. A low $CV_i$ indicates that removing term $S_i$ causes little instability in the rest of the model, suggesting it is a candidate for pruning. MSSR iteratively removes the term with the smallest coefficient of variation (CV) and recalculates the CVs for all remaining terms after each deletion. The pruning process stops once the $R^2$ (La Cava et al., 2021) score of the expression drops below a predefined threshold. This procedure allows MSSR to effectively identify and remove redundant terms that may arise from overfitting, thereby simplifying the expression.

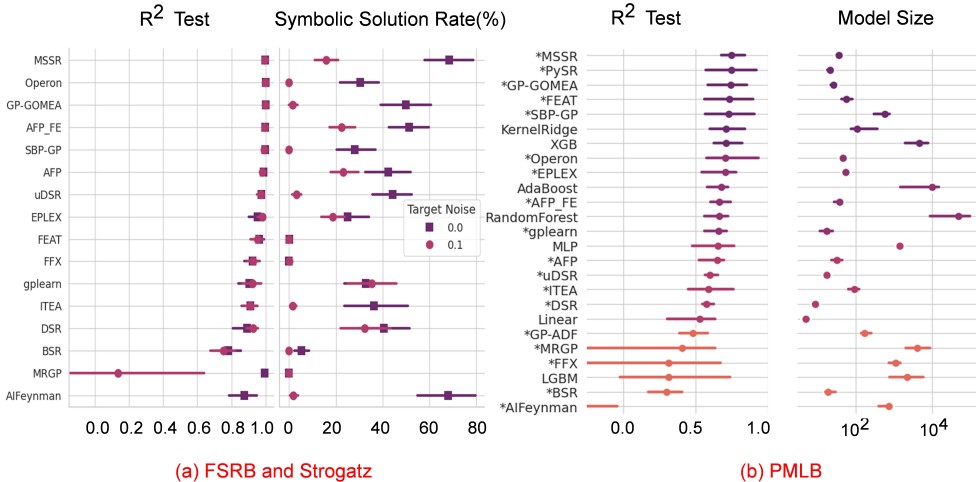

Figure 2: Result comparisons on SRBench.

# 4 EXPERIMENTS

## 4.1 SRBENCH

To verify MSSR's performance in symbolic regression, we conducted experiments using SRBench (La Cava et al., 2021), which contains three benchmark datasets: the Feynman Symbolic Regression Benchmarks (FSRB) (Udrescu & Tegmark, 2020), Strogatz Benchmarks (La Cava et al., 2016), and Penn Machine Learning Benchmarks (PMLB) (Olson et al., 2017). For the PMLB dataset collection, we ran MSSR 10 times on each dataset with different random seeds and compared it with 24 baseline algorithms (from SRbench, uDSR(Landajuela et al., 2022), PySR(Cranmer, 2023) and GP-ADF[2]) using two key metrics: $R^2$ and model size (the number of nodes in the expression). For the FSRB and Strogatz benchmarks, we independently ran MSSR 20 times on each dataset (10 runs without noise and 10 with Gaussian noise). We then compared its performance against 14 baseline algorithms based on three metrics: $R^2$, model size, and symbolic recovery rate (La Cava et al., 2021). Details regarding parameters and datasets are provided in the Appendix A.2.

## 4.1.1 RESULTS

The performance of MSSR compared to the baseline SR methods is summarized in Fig. 2. As shown in Fig. 2(a) for the FSRB and Strogatz benchmarks, MSSR achieves more accurate results

---

[2]https://github.com/DEAP/deap

(in terms of $R^2$ score) than most baseline methods and maintains robust performance at a noise level of $0.1$. Considering both noise-free and $0.1$ noise level settings, MSSR achieves the highest overall symbolic recovery rate of 84.21% (68.39% from noise-free and 15.82% from noisy cases). The strong performance of MSSR can be attributed to its discovery of valuable sub-expressions during the search. Identifying building blocks that closely match parts of the correct expression significantly increases the symbolic recovery rate and overall accuracy. Moreover, our use of the noise-resistant mutual information metric to evaluate sub-expressions enables MSSR to maintain this stability in the presence of noise.

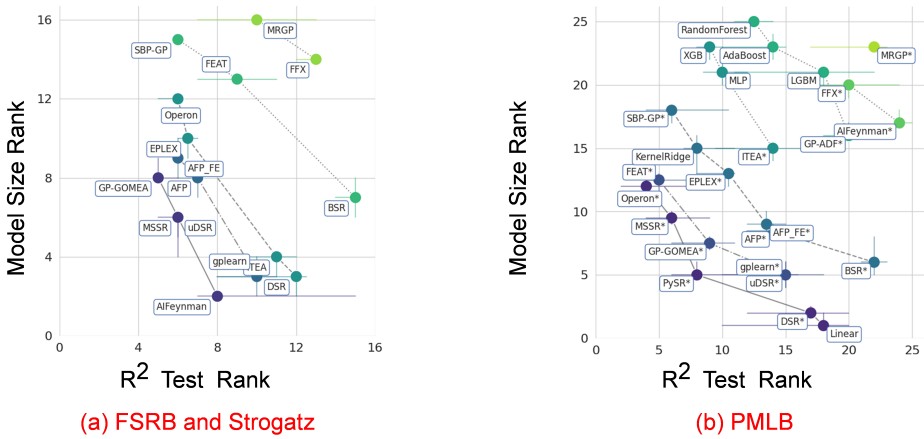

(a) FSRB and Strogatz          (b) PMLB

Figure 3: Pareto front plot based on model size rank and $R^2$ score rank.

Recovering a ground truth expression under noisy conditions is challenging. Nevertheless, some methods, such as gplearn[3] and DSR (Petersen et al., 2019), still perform very well. Although most methods suffer a significant drop in symbolic recovery rate, often approaching zero, MSSR consistently discovers the correct expressions. This resilience is primarily attributed to its pruning mechanism, which effectively filters out redundant terms and increases the likelihood of identifying the ground truth expression.

On the PMLB benchmark, the comparison with 24 baseline methods is shown in Fig. 2(b). MSSR demonstrates competitive performance on these high-dimensional datasets, achieving an $R^2$ score that ranks among the top. Compared with the highly competitive PySR(Cranmer, 2023), MSSR achieves a slightly higher $R^2$ (0.7497) and demonstrates better stability. Furthermore, among the top 25% of methods with the highest $R^2$ score, MSSR has the third smallest model size. This balance can be attributed to how MSSR constrains complexity: it limits the number of terms in the final sum and indirectly restricts the complexity of the sub-expressions within the library. Therefore, MSSR avoids the problem of generating excessively large mathematical expressions.

A Pareto front analysis, shown in Fig. 3, further evaluates the balance between the $R^2$ score and model size. MSSR consistently ranks on the optimal Pareto front across the Feynman, Strogatz, and PMLB benchmarks, indicating that it is among the most effective methods for achieving a state-of-the-art trade-off between model accuracy and structural simplicity.

### 4.1.2 ABLATION STUDY

To evaluate the contribution of each component in MSSR, we conduct an ablation experiment. MSSR comprises three key components: policy networks, a sub-expression library, and evolutionary mechanisms. As shown in Table. 1, the combined effect of these components results in great performance. Further ablation experiments on pruning strategies are presented in the Appendix A.4.

---

[3]https://github.com/trevorstephens/gplearn

Table 1: Ablation results. "Without Policy Network" indicates that variables are sampled according to mutual information, and operators are sampled randomly.

| Setting | Benchmark | A | B | C | D | E |
|---|---|---|---|---|---|---|
| Policy Network | | ✓ | ✓ | | ✓ | |
| Sub-expression Library | | ✓ | ✓ | ✓ | | ✓ |
| Evolution | | ✓ | | ✓ | | |
| Dataset | | $R^2$ Test | | | | |
| feynman_III_15_12 | FSRB | **1.0** | 0.7712 | 0.9997 | 0.2689 | 0.2404 |
| feynman_I_12_2 | FSRB | **1.0** | 0.9999 | 0.8763 | 0.6783 | 0.2852 |
| feynman_III_19_51 | FSRB | **0.9823** | 0.8227 | 0.8749 | 0.2338 | 0.4666 |
| strogatz_vdp1 | Strogatz | **0.9973** | 0.9953 | 0.8210 | 0.3268 | 0.2842 |
| 596_fri_c2_250_5 | PMLB | **0.8498** | 0.7973 | 0.6889 | 0.1683 | 0.6374 |
| **Average** | | **0.9658** | 0.8772 | 0.8521 | 0.3352 | 0.3827 |

## 4.2 PDE DISCOVERY

For parametric partial differential equations (PDEs), we consider the following form (Rudy et al., 2019):

$$u_t = f(u, u_x, u_{xx}, \cdots, g(\zeta)) = \sum_{i=0}^{n} f_i(u, u_x, u_{xx}, \cdots) \times g_i(\zeta) \qquad (8)$$

where $u_t, u_x, \cdots$ are partial derivatives, $f_i$ are functions of the state variable $u$ and its spatial derivatives, and $g_i(\zeta)$ is a coefficient function with $\zeta \in \{t, x\}$, i.e., $g_i$ depends solely on either $t$ or $x$.

To find a general PDE that fits multiple datasets, the goal is to identify a common structure $\{f_i\}$ that can be adjusted by the coefficients $\{g_i\}$ for each dataset. Accordingly, each dataset is partitioned along both $t$ and $x$; focusing on temporal partitions, for example, yields subdatasets $\{D_j^k | j = 1, 2, \cdots\}$. MSSR is then applied to sample a general structure $f = \{f_i \mid i = 1, 2, \cdots\}$ (without constants), which is shared across all datasets. Each sub-expression $f_i$ in this structure is multiplied by a coefficient $c_{ij}^k$ for each subdataset $D_j^k$. These coefficients are found via constant optimization, and a coefficient of determination $R_t^2$ is computed. A similar procedure is applied to spatial partitions to yield $R_x^2$. The coefficient sequence for each term is then associated with the variable ($t$ or $x$) that yields the higher $R^2$.

We define a coefficient sequence $\xi_i^k = \{c_{ij}^k \mid j = 1, 2, \cdots\}$ for the $i$-th sub-expression across the $k$-th dataset. For each sequence $\xi_i^k$, we evaluate its mean and variance. If a sequence $\xi_i^k$ exhibits small variance, the corresponding coefficient function $g_i$ is treated as a constant; otherwise, DSR (Petersen et al., 2019) is used to model it as a function of the independent variable ($t$ or $x$). Furthermore, sequences $\xi_i^k$ with low mean values are considered insignificant. To validate this approach, we constructed three datasets for the Burgers' equation, each with different parameters and initial conditions, based on the procedure described by (Rudy et al., 2019) and (Xu et al., 2021). The information of the three datasets is presented in the Appendix A.3.

### 4.2.1 RESULTS

For the three Burgers datasets, Fig. 4 illustrates the true values of $u$, the solution of the discovered PDE, and their absolute differences. The corresponding true and discovered PDEs for each dataset are also shown. The general structure found by MSSR is highlighted in red. It is evident that the PDEs discovered for each dataset share a common structure, consistently including the differential terms '$u \times u_x$' and '$u_{xx}$'. Since this structure matches that of the true PDEs, it suggests MSSR has identified a generalizable model capable of describing different datasets. This capability is likely due to MSSR's search strategy, which proceeds from simple to complex sub-expressions and is thus well-suited for discovering the concise forms of PDEs.

After obtaining the common skeleton using MSSR, we applied a symbolic regression method (DSR (Petersen et al., 2019)) to find closed-form expressions for the numerical coefficient sequences. This process yielded expressions such as '$-0.999 - 0.249 \times sin(t)$'. Although these discovered coeffi-

cient expressions differ from the true ones, the solution of the final discovered PDE still accurately depicts the fluid flow, as shown by the small absolute differences in Fig. 4.

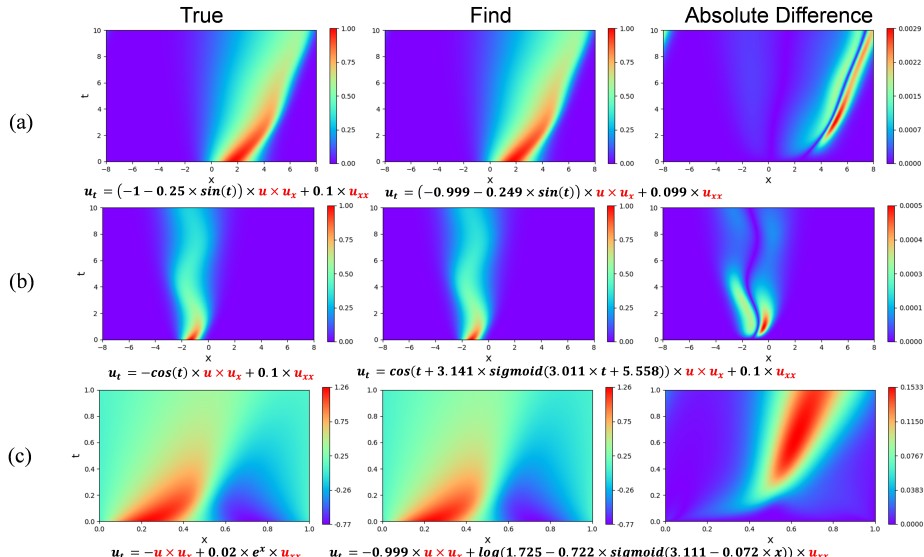

Figure 4: Actual values of $u$, the values of $u$ obtained by solving the found PDE, and absolute differences on Burgers datasets.

# 5 DISCUSSION

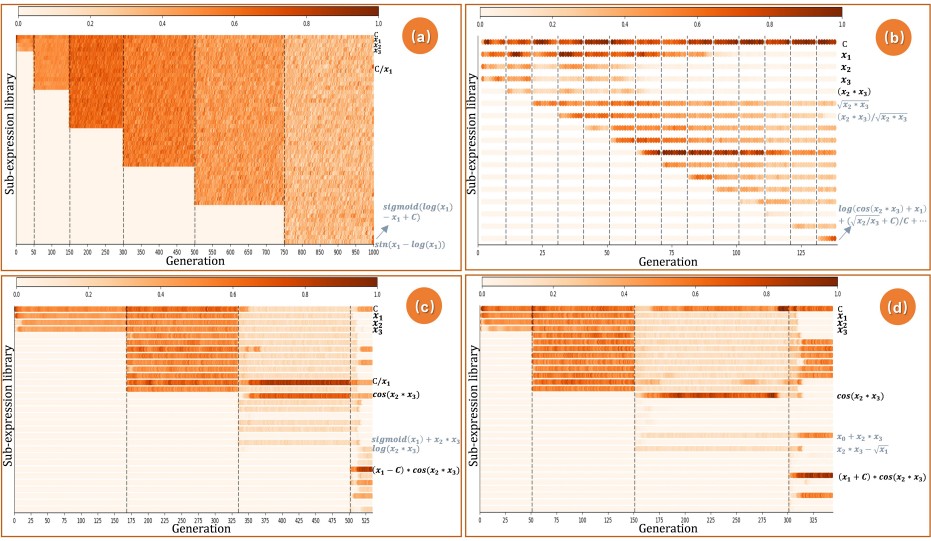

Figure 5: Heat map of selection frequencies in sub-expression library on "$y = 2 \times x_1 \times (1 - cos(x_2 \times x_3))$" with different ablation and parameter settings using MSSR. (a) represents MSSR without evolution. (b) represents MSSR with more frequent sub-expression library updates. (c) represents MSSR without a gradually increasing update interval for the sub-expression library. (d) represents MSSR.

The strong performance of MSSR is mainly due to its ability to discover valuable sub-expressions that improve accuracy. Figure 5 shows the selection frequency of sub-expressions from the library $\mathcal{L}$ across generations for the benchmark '$y = 2 \times x_1 \times (1 - cos(x_2 \times x_3))$'. In these plots, correct sub-expressions are shown in black and incorrect ones in gray. Without the evolutionary strategy

(Fig. 5(a)), the selection probabilities are nearly uniform, indicating limited sub-expression diversity and fewer discoveries of correct components. In contrast, the full MSSR model uses evolution to enrich this diversity and increase the chance of identifying the correct expressions. The update frequency of the library is also critical. Figure 5(b) shows that updating $\mathcal{L}$ too frequently results in overly complex, less useful sub-expressions. Although the model may reach the $R^2$ threshold early, valuable sub-structures are often missed. MSSR avoids this by limiting the update frequency, which helps keep the library's sub-expressions compact and useful.

Furthermore, using a fixed update interval (Fig. 5(c)) is less efficient, even though correct sub-expressions like '$cos(x_2 \times x_3)$' are eventually found. MSSR, by gradually increasing the update interval (Fig. 5(d)), finds the correct expressions in fewer generations. Early updates tend to yield little gain when the library contains low-quality expressions; the adaptive schedule improves efficiency by focusing updates where they matter most.

Nevertheless, this work has two main limitations. First, the limits on the number of sub-expressions and the frequency of library updates may need to be tuned for different datasets. Second, as the sub-expression library undergoes multiple updates, it may accumulate increasingly complex nested expressions. When this complexity exceeds that of the ground-truth solution for a given dataset, the generated expressions may become unnecessarily redundant.

Our future work will focus on integrating various symbolic regression algorithms to mine more meaningful and informative sub-expression libraries. In addition, we will pay particular attention to balancing the efficiency of generating sub-expressions with their overall effectiveness.

## 6 CONCLUSION

In this study, we introduced MSSR, a novel method for symbolic regression designed to tackle the fundamental challenge of searching an immense expression space. Our framework is built on the core principle that discovering and reusing valuable sub-expressions is a more efficient search strategy than constructing solutions from atomic operators. MSSR employs a cooperative Multi-Agent reinforcement learning framework to sample expressions from its sub-expression library. To sample more valuable sub-expressions, MSSR optimizes each agent's policy network using a custom policy gradient. In parallel, MSSR evolves and mines valuable sub-expressions to update the library. Finally, a CV-based pruning technique simplifies the discovered expressions to ensure parsimony. Compared with 24 baseline methods, MSSR achieves a state-of-the-art balance between accuracy and simplicity. Additionally, MSSR successfully identified the general structure of a PDE from a physical dataset. By shifting the focus from atomic operators to meaningful sub-expressions, MSSR provides a powerful framework for constraining the combinatorial search in symbolic regression, enabling the discovery of more accurate and parsimonious models from complex data.

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

# A  APPENDIX

## A.1  ALGORITHM

The pseudocode of MSSR is presented in Algorithm 1. It mainly includes four parts: sampling complete expressions based on multi-agents, training the policy networks of multi-agents, updating the sub-expression library and pruning the expression.

**Algorithm 1** MSSR algorithm

---

**Input**:dataset $D = \{X_t, y_t\}$,max_epoch,max_step,batch size $N$,learning rate $\alpha$; entropy coeficient $\lambda_{\mathcal{H}}$; risk factor $\varepsilon$; reward function $R$

**Output**:best expression $f^*$

1: Initialize policy networks $N_l, N_o, N_r$ with $\theta_l, \theta_o, \theta_r$
2: **Function** sampling(max_step):
3:    $\tau \leftarrow []$
4:    $C \leftarrow []$
5:    **for** 1 to max_step **do**
6:       $\pi_l \leftarrow N_l(y; \theta_l)$
7:       $l \leftarrow \text{Categorical}(\pi_l)$
8:       $\tau \leftarrow \tau || l$
9:       $\pi_r \leftarrow N_o(y, l; \theta_o)$
10:      $o \leftarrow \text{Categorical}(\pi_o)$
11:      $\tau \leftarrow \tau || o$
12:      $\pi_r \leftarrow N_r(y, l, o; \theta_r)$
13:      $r \leftarrow \text{Categorical}(\pi_r)$
14:      $\tau \leftarrow \tau || r$
15:      $y', C \leftarrow \text{OptimizeConstant}(\tau, C)$ /* Warm-start from previous $C$ for optimization*/
16:      $y \leftarrow y - y'$
17:    **end for**
18:    **return** $\tau$
19: **for** 1 to max_epoch **do**
20:    $f \leftarrow \{\text{sampling(max\_step)}\}_{i=1}^{N}$
21:    $R \leftarrow \{R_{f_i}\}_{i=1}^{N}$
22:    $R_\varepsilon \leftarrow (1-\varepsilon)$-quantile of $R$
23:    $f_\varepsilon \leftarrow \{f_i : R_{f_i} \geq R_\varepsilon\}$
24:    $R \leftarrow \{R_{f_i} : R_{f_i} \geq R_\varepsilon\}$
25:    **for** $i \in \{l, o, r\}$ **do**
26:      $\hat{g}_1 \leftarrow \text{ReduceMean}\big((\mathcal{R} - R_\varepsilon)\nabla_{\theta_i} \log p(f_{\varepsilon_i}|\theta_i)\big)$
27:      $\hat{g}_2 \leftarrow \text{ReduceMean}\big(\lambda_{\mathcal{H}} \nabla_{\theta_i} \mathcal{H}(f_{\varepsilon_i}|\theta_i)\big)$
28:      $\theta \leftarrow \theta + \alpha(\hat{g}_1 + \hat{g}_2)$
29:    **end for**
30:    **if** $\max \mathcal{R} > R_{f^\star}$ **then**
31:      $f^\star \leftarrow f^{(\arg\max \mathcal{R})}$
32:    **end if**
33:    **if** time to update sub-expression library $\mathcal{L}$ **then**
34:      $f_{GP}^{(0)} \leftarrow f_\varepsilon$
35:      $f_{GP} = \text{GP}(f_{GP}^{(0)})$
36:      sub_exprs $\leftarrow f_\varepsilon + f_{GP}$ /*$f_\varepsilon$ split by steps; $f_{GP}$ split by subtree height*/
37:      $C \leftarrow \text{mutual(sub\_exprs)}$
38:      $\mathcal{L} \leftarrow \text{update}(\mathcal{L}, \text{sub\_exprs}, C)$
39:      max_step $\leftarrow$ max_step - 1
40:    **end if**
41: **end for**
42: $f^* \leftarrow \text{prune}(f^*)$
43: **return** $f^*$

---

Table 2: PMLB datasets

| 1029_LEV | 1096_FacultySalaries | 192_vineyard | 210_cloud |
|---|---|---|---|
| 225_puma8NH | 601_fri_c1_250_5 | 656_fri_c1_100_5 | 687_sleuth_ex1605 |
| 228_elusage | 547_no2 | 595_fri_c0_1000_10 | 596_fri_c2_250_5 |
| 609_fri_c0_1000_5 | 611_friH_c3_100_5 | 612_fri_c1_1000_5 | 628_fri_c3_1000_5 |
| 665_sleuth_case2002 | 666_rmftsa_ladata | 594_fri_c2_100_5 | 712_chscase_geyser1 |
| 579_fri_c0_250_5 | 542_pollution | 659_sleuth_ex1714 | 706_sleuth_case1202 |
| 591_fri_c1_100_10 | | | |

## A.2 EXPERIMENTAL SETUPS FOR SRBENCH

In the experiment, PMLB includes 25 black-box datasets, as shown in Table 2. Meanwhile, FSRB and Strogatz benchmarks include 55 datasets, as shown in Table 3. The detailed parameters for MSSR are provided in Table 4, while the parameters for the baseline algorithms are taken from the SRBench study (La Cava et al., 2021).

Table 3: FSRB and Strogatz datasets

| feynman_III_19_51 | feynman_III_15_12 | feynman_I_43_16 | feynman_I_43_31 |
|---|---|---|---|
| feynman_I_39_22 | feynman_I_47_23 | feynman_II_8_31 | feynman_II_3_24 |
| feynman_II_37_1 | feynman_II_27_18 | feynman_II_27_16 | feynman_I_39_1 |
| feynman_II_15_4 | feynman_III_15_14 | feynman_I_14_4 | feynman_II_34_29a |
| feynman_II_34_2 | feynman_II_15_5 | feynman_I_43_43 | feynman_I_18_14 |
| feynman_I_34_8 | feynman_I_12_11 | feynman_I_11_19 | feynman_I_32_5 |
| feynman_I_12_2 | feynman_II_38_3 | feynman_II_38_14 | feynman_III_12_43 |
| feynman_II_11_28 | feynman_II_2_42 | feynman_III_17_37 | feynman_II_11_3 |
| feynman_I_14_3 | feynman_I_12_5 | feynman_I_25_13 | feynman_I_12_1 |
| feynman_I_29_4 | feynman_I_34_27 | feynman_I_18_12 | feynman_III_15_27 |
| feynman_II_13_17 | strogatz_lv1 | strogatz_vdp1 | strogatz_vdp2 |
| strogatz_predprey1 | strogatz_predprey2 | strogatz_bacres1 | strogatz_bacres2 |
| strogatz_glider1 | strogatz_glider2 | strogatz_shearflow1 | strogatz_shearflow2 |
| strogatz_barmag1 | strogatz_barmag2 | strogatz_lv2 | |

Table 4: MSSR Parameter Setting

| Parameter | Value | Parameter | Value |
|---|---|---|---|
| Function Set | $+,-,\times,\div,sin,cos,log,sqrt,sigmoid,id$ | | |
| **Parameter** | **Value** | **Parameter** | **Value** |
| Max Generations | 1000 | Population Size | 200 |
| Max Terms | 6 | Risk Threshold | 0.95 |
| Evo-Population Size | 200 | Entropy Coefficient | 0.005 |
| Evo-Max Generations | 20 | $R^2$ Stopping Threshold | 0.99999 |
| Evo-Crossover Probability | 0.5 | Sparse Threshold | 0.01 |
| Evo-Mutation Probability | 0.5 | Updating Generations | 50,100,150,200,250 |

## A.3 EXPERIMENTAL SETUPS FOR PDE DISCOVERY

The parameters of the three datasets in the PDE discovery experiment are shown in Table. 5.

## A.4 ABLATION STUDY

We additionally validated the effectiveness of the pruning strategy, as shown in Table. 6. It can be seen that it is very effective for the FSRB benchmark, reducing the model size while ensuring that $R^2$ does not decrease on the test dataset. The effect on other benchmarks (Strogatz and PMLB) is not very significant.

Table 5: Experimental Setups

| Setup | Initial Condition ($u_0$) | $x, t$ range |
|-------|---------------------------|--------------|
| a | $u_0 = e^{-(0.6x-1)^2}$ | $x \in [-8, 8], t \in [0, 10]$ |
| b | $u_0 = e^{-(1.5x+2)^2}$ | $x \in [-8, 8], t \in [0, 10]$ |
| c | $u_0 = (1.5 - x) \times sin(6.28 \times x)$ | $x \in [0, 1], t \in [0, 1]$ |

Table 6: Pruning results.

| | Before Pruning | | After Pruning | |
|---|---|---|---|---|
| | $R^2$ Test | Model Size | $R^2$ Test | Model Size |
| FSRB (no noisy) | 0.9992 | 14.4878 | 0.9992 | 12.9902 |
| FSRB (0.1 noisy) | 0.9933 | 33.4414 | 0.9930 | 27.0341 |
| Strogatz (no noisy) | 0.9792 | 33.9000 | 0.9789 | 32.9642 |
| Strogatz (0.1 noisy) | 0.9762 | 40.1928 | 0.9742 | 38.5785 |
| PMLB | 0.6615 | 38.4560 | 0.6615 | 37.8000 |
| **Average** | **0.9218** | **32.0956** | **0.9213** | **29.8734** |

A.5 PROOF OF AGENT GRADIENT

**Proposition 1** (Agent Gradient). *The gradient of the cooperative multi-agent objective is given by*

$$\nabla_{\theta_i} J_{risk}(\theta) \approx \frac{1}{\varepsilon N} \sum_{k=1}^{N} \left[ R_{f^{(k)}} - \tilde{R}_\varepsilon \right] \cdot \mathbf{1}_{R_{f^{(k)}} \geq \tilde{R}_\varepsilon} \nabla_{\theta^i} \log p(f_i^{(k)} \mid \theta_i) \quad (9)$$

**Proof.** First, let $p_i(f_i^{(k)})$ denote the probability of generating the part of the $k$-th expression $f^{(k)}$ in the policy network parameterized by $\theta_i$ in the $i$-th agent, where i represents one of $l$, $o$ or $r$. The $p_i(f_i^{(k)})$ decomposes into the product of step-wise selection probabilities in $n$ time steps as following function:

$$p_l(f_l^{(k)} \mid \theta_l) = \prod_{t=1}^{n} \pi_{l_t}(a_{l_t}^{(k)} \mid s_{l_t}^{(k)}; \theta_l) \quad (10)$$

$$p_o(f_o^{(k)} \mid \theta_o) = \prod_{t=1}^{n} \pi_{o_t}(a_{o_t}^{(k)} \mid s_{o_t}^{(k)}; \theta_o) \quad (11)$$

$$p_r(f_r^{(k)} \mid \theta_r) = \prod_{t=1}^{n} \pi_{r_t}(a_{r_t}^{(k)} \mid s_{r_t}^{(k)}; \theta_r) \quad (12)$$

Then, the sequence of actions of all agents that generates the overall probability of the expression $f^{(k)}$ is defined as follows:

$$p(f^{(k)} \mid \theta) = p_l(f_l^{(k)} \mid \theta_l) \cdot p_o(f_o^{(k)} \mid \theta_o) \cdot p_r(f_r^{(k)} \mid \theta_r) \quad (13)$$

In order to achieve the goal, combined with Monte Carlo gradient estimation and risk-seeking policy gradient(Petersen et al., 2019), the gradient can be described as follows:

$$\nabla_\theta J_{\text{risk}}(\theta) \approx \frac{1}{\varepsilon N} \sum_{k=1}^{N} \left[ R_{f^{(k)}} - \tilde{R}_\varepsilon \right] \cdot \mathbf{1}_{R_{f^{(k)}} \geq \tilde{R}_\varepsilon} \nabla_\theta \log p(f^{(k)} \mid \theta) \quad (14)$$

where $\epsilon$ represents the probability threshold of the quantile.

Logarithmic probability can be decomposed as follows.

$$log(p(f^{(k)} \mid \theta)) = log(p_l(f_l^{(k)} \mid \theta_l)) + log(p_o(f_o^{(k)} \mid \theta_o)) + log(p_r(f_r^{(k)} \mid \theta_r)) \quad (15)$$

When the gradient is calculated for the parameter $\theta_l$ of the policy network in agent $N_l$, only $log(p_l(f_l^{(k)} \mid \theta_l))$ produces a nonzero gradient:

$$\nabla_{\theta_l} log(p(f^{(k)} \mid \theta)) = \nabla_{\theta_l} log(p_l(f_l^{(k)} \mid \theta_l)) \tag{16}$$

, which is the same as $N_o$ and $N_r$. Finally, the gradient for each policy network can be redefined as:

$$\nabla_{\theta_l} J_{\text{risk}}(\theta) \approx \frac{1}{\varepsilon N} \sum_{k=1}^{N} \left[ R_{f^{(k)}} - \tilde{R}_\varepsilon \right] \cdot \mathbf{1}_{R_{f^{(k)}} \geq \tilde{R}_\varepsilon} \nabla_{\theta_l} \log(p_l(f_l^{(k)} \mid \theta_l)) \tag{17}$$

$$\nabla_{\theta_o} J_{\text{risk}}(\theta) \approx \frac{1}{\varepsilon N} \sum_{k=1}^{N} \left[ R_{f^{(k)}} - \tilde{R}_\varepsilon \right] \cdot \mathbf{1}_{R_{f^{(k)}} \geq \tilde{R}_\varepsilon} \nabla_{\theta_o} \log(p_o(f_o^{(k)} \mid \theta_o)) \tag{18}$$

$$\nabla_{\theta_r} J_{\text{risk}}(\theta) \approx \frac{1}{\varepsilon N} \sum_{k=1}^{N} \left[ R_{f^{(k)}} - \tilde{R}_\varepsilon \right] \cdot \mathbf{1}_{R_{f^{(k)}} \geq \tilde{R}_\varepsilon} \nabla_{\theta_r} \log(p_r(f_r^{(k)} \mid \theta_r)) \tag{19}$$

### A.6 Algorithm Performance on Actual Results

Fig. 6. presents the pareto plot based on the actual values ($R^2$ Test and model size) (Fong & Motani), where points closer to the top-left corner indicate better performance. Across all benchmarks, MSSR consistently ranks among the methods positioned near the top-left corner.

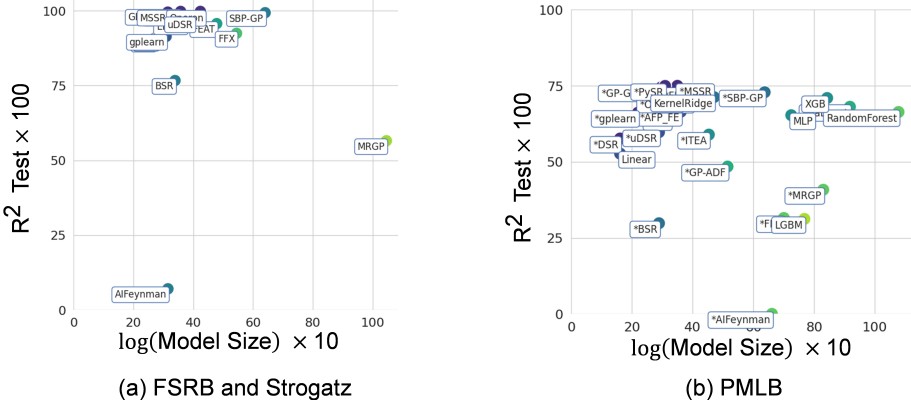

(a) FSRB and Strogatz

(b) PMLB

Figure 6: Pareto front plot based on model size and $R^2$ score.

