# OpenReview forum: "Mining Valuable Sub-Expressions for Symbolic Regression"
_ICLR.cc/2026/Conference — Submitted to ICLR 2026_

### Official Review · Reviewer_5Pwr · 2025-10-28

**Soundness:** 1
**Presentation:** 3
**Contribution:** 2
**Rating:** 2
**Confidence:** 4

**Summary:**

This paper introduces MSSR, a novel framework for symbolic regression that leverages sub-expressions within a multi-agent reinforcement learning framework. While the idea of reusing sub-expressions is promising, the manuscript has several limitations in methodological justification, experimental rigor, and contextualization within the existing literature. Below are detailed comments and suggestions for improvement.

**Strengths:**

**Robust and Adaptive Library Management​**

​Dynamic Library Updates: The sub-expression library Lis not static; it evolves using GP based on mutual information (MI) with the target. This ensures the library remains relevant and enriched with components that have high predictive utility, especially under noisy conditions.

​Information-Theoretic Sub-expression Evaluation: Using MI to evaluate sub-expressions provides a noise-resistant measure of relevance, contributing to the method's robustness in noisy data scenarios.

**Weaknesses:**

**1. Limitations in Expression Form Assumption​**

The proposed method inherently assumes that target expressions can be decomposed into a weighted sum of sub-expressions. This design may favor expressions naturally adhering to this form (e.g., additive models) but could struggle with expressions that do not decompose additively. In such cases, MSSR might overcomplicate the solution by forcing a sum-of-terms structure, potentially leading to less parsimonious fits than methods without this structural bias.

**2. Insufficient Coverage of Related Work​**

The literature review lacks depth in several key areas:

​GP-based methods with similar ideas: Techniques like Genetic Programming with Automatically Defined Functions (GP-ADF) or modular GP explicitly evolve and reuse sub-expressions but are not discussed.

​RL-based symbolic regression methods: Prior works such as GP-RL or hierarchical RL approaches for expression construction are not adequately compared.

**3. Methodological Clarifications Needed​**

​Sub-expression selection via Mutual Information (MI)​: The manuscript mentions using MI to evaluate sub-expressions but omits critical details: How is MI computed between a sub-expression and the target? What is the exact formulation? Are continuous outputs discretized? Clarification is essential for reproducibility.

​Agent coordination and scalability: The framework employs 3 agents per sub-expression, implying 3×n agents for an expression with n terms. It is unclear how the search space scales with term count or how inter-term dependencies are handled. Since agents update independently, the approach may overlook synergies between terms, potentially hindering global optimization.

**​4. Experimental Comparisons and Baselines​**

​Comparison with state-of-the-art: The omission of PySR (a recently published and widely recognized SR tool) undermines the credibility of claimed advancements. PySR should be included in benchmarks.

​Ablation study limitations: The ablation experiments are limited to a few datasets, which may coincidentally align with the method’s structural assumptions. To demonstrate generalizability, ablations should cover all benchmark categories (PMLB, FSRB, Strogatz).

​Lack of comparison with GP-ADF and pretrained methods: GP-ADF directly addresses sub-expression reuse and should be included. Additionally, pretrained approaches like SNIP or NeSymReS, which excel in model complexity reduction, are not evaluated.

**Questions:**

The questions are described in the "Weakness" section.

---

> ### Author Response · Authors · 2025-11-20
>
> We are grateful to the reviewer 5Pwr for providing valuable comments and constructive suggestions. Below are our responses to your questions respectively.
>
> **Weaknesses**
> **W1**: **Limitations in Expression Form Assumption**
> **W1.1**: The proposed method inherently assumes that target expressions can be decomposed into a weighted sum of sub-expressions. This design may favor expressions naturally adhering to this form (e.g., additive models) but could struggle with expressions that do not decompose additively. In such cases, MSSR might overcomplicate the solution by forcing a sum-of-terms structure, potentially leading to less parsimonious fits than methods without this structural bias.
> **R1.1**: In this work, expressions can be understood as having a two-layer nested structure. The top-level operator is a sum, and each term is a sub-expression $(S_l,O,S_r)$. Within a sub-expression, the left and right operands can themselves be sub-expressions, and the operator is chosen from a predefined operator set $\lbrace  +, -, \times, /, \sin, \cos, \log, \dots \rbrace $. For example, consider the target expression $\frac{1.1x_1 + 1.2x_2}{\sin(x_3)}$ The sub-expression library contains $1.1x_1 + 1.2x_2$ and $sin⁡(x_3)$. The algorithm utilizes agent $N_l$ to sample $1.1x_1 + 1.2x_2$ as the left  sub-expression from the sub-expression library, agent $N_o$ to sample '/' from the operator set, and agent $N_r$ to sample $sin⁡(x_3)$ as the right  sub-expression from the sub-expression library to construct the target expression. So, after combining the three sampling results, MSSR can get the sub-expression $\frac{1.1x_1 + 1.2x_2}{\sin(x_3)}$. As the fitness of the sub-expression satisfies the stop threshold, MSSR stops and cannot use the top-level operator - sum. Therefore, MSSR can handle the solution that cannot be decomposed by addition. This means that MSSR uses the three agents above to generate sub-expressions to obtain the solution.
> For example, as in Feynman_III_15_12, MSSR can recover the solution $y = 2 \times x_1 \times \left(1 - \cos(x_2 \times x_3)\right)$. The detailed process of finding the right solution is as follows. MSSR uses the three agents to find the sub-expressions $1 - \cos(x_2 \times x_3)$ and $2 \times x_1$ with higher mutual information (MI), which are saved into the sub-expression library. Subsequently, by sampling these two sub-expressions from the library and '$\times$' from the operator set, the ground-truth expression can be derived.
>
> **W2:** **Insufficient Coverage of Related Work**
> **W2.1**: GP-based methods with similar ideas: Techniques like Genetic Programming with Automatically Defined Functions (GP-ADF) or modular GP explicitly evolve and reuse sub-expressions but are not discussed.
> **R2.1**: We provided a more comprehensive discussion of related works in Section 2. Specifically, we now include GP-based methods that explicitly evolve and reuse sub-expressions, such as GP-ADF[1], GEP-ADF[2], SL-GEP[3].
>
> The clarified content in the Section 2 of the modified manuscript is listed as follows.
>
> ADF is used to achieve code reuse and modular development, similar to the concept of reusable sub-expressions[2].
> Some ADF-based methods may lead to multi-layer nesting [2,3], resulting in structural redundancy. However, MSSR algorithm avoids this defect by focusing on mining increasingly meaningful sub-expressions with high mutual information.
> [1] https://github.com/DEAP/deap
> [2] Ferreira, Cândida. "Automatically defined functions in gene expression programming." Genetic Systems Programming: Theory and Experiences. Berlin, Heidelberg: Springer Berlin Heidelberg, 2006. 21-56.
> [3] Zhong, Jinghui, Yew-Soon Ong, and Wentong Cai. "Self-learning gene expression programming." IEEE Transactions on Evolutionary Computation 20.1 (2015): 65-80.

---

> ### Author Response · Authors · 2025-11-20
>
> We are grateful to the reviewer 5Pwr for providing valuable comments and constructive suggestions. This is another part of our response.
>
> **W2.2**: RL-based symbolic regression methods: Prior works such as GP-RL or hierarchical RL approaches for expression construction are not adequately compared.
> **R2.2**: We have added the relevant GP-RL symbolic regression literature to Section 2 and provided more detailed descriptions of these methods.
>
> The clarified content in the Section 2 of the modified manuscript is listed as follows.
>
> For example, DSR-GP [1] and uDSR [2] exemplify the first category by using a neural policy to generate a high-quality initial population for a subsequent GP search. Similarly, RSRM [3], which is based on MCTS and double Q-learning, also leverages evolutionary principles, applying GP to optimize expression trees generated by a separate process.
>
> [1] Mundhenk, T. Nathan, et al. "Symbolic regression via neural-guided genetic programming population seeding." arXiv preprint arXiv:2111.00053 (2021).
> [2] Landajuela, Mikel, et al. "A unified framework for deep symbolic regression." Advances in Neural Information Processing Systems 35 (2022): 33985-33998.
> [3] Xu, Yilong, Yang Liu, and Hao Sun. "Reinforcement symbolic regression machine." The Twelfth International Conference on Learning Representations. 2024.
>
>
> **W3**: **Methodological Clarifications Needed**
> **W3.1**: Sub-expression selection via Mutual Information (MI): The manuscript mentions using MI to evaluate sub-expressions but omits critical details: How is MI computed between a sub-expression and the target? What is the exact formulation? Are continuous outputs discretized? Clarification is essential for reproducibility.
> **R3.1**: MI is used to evaluate the mutual information between the value X of a sub-expression and the target value Y, which defined as $I(X; Y) = \sum_{y \in Y} \sum_{x \in X} p(x, y) \log\left( \frac{p(x, y)}{p(x)p(y)} \right)$. X and Y are continuous random variables, and the mutual information is estimated using a histogram-based method [1].
>
> [1] Zojaji, Zahra, and Mohammad Mehdi Ebadzadeh. "Semantic schema theory for genetic programming." Applied Intelligence 44.1 (2016): 67-87.
>
> **W3.2**: Agent coordination and scalability: The framework employs 3 agents per sub-expression, implying 3×n agents for an expression with n terms. It is unclear how the search space scales with term count or how inter-term dependencies are handled. Since agents update independently, the approach may overlook synergies between terms, potentially hindering global optimization.
> **R3.2**: In our framework, the three agents are responsible for generating each sub-expression, corresponding to the left sub-expression, the operator, and the right sub-expression. Therefore, for an expression with n sub-expressions, only three agents are employed at a time for generating each sub-expression, rather than 3×n agents. During training, all three agents share the same reward value, which encourages coordinated construction of correct expressions. For specific examples, please see the response to R1.1 above.
>
> **W4**: **Experimental Comparisons and Baselines**
>
> **W4.1**: Comparison with state-of-the-art: The omission of PySR (a recently published and widely recognized SR tool) undermines the credibility of claimed advancements. PySR should be included in benchmarks.
> **R4.1**: We have tested PySR[1] on the PMLB benchmark to evaluate its performance, particularly in high-dimensional settings. Compared with PySR, MSSR achieves a slightly higher R², which may be attributed to its continuous mining of meaningful sub-expressions to construct high-quality expressions. PySR sets a limit on expression length, whereas MSSR does not. As a result, the solutions obtained by MSSR tend to be longer than those from PySR. This is because MSSR needs to be compared with high-accuracy algorithms such as Operon, which do not impose length constraints. Moreover, MSSR’s ability to reuse sub-expressions can help reduce the overall solution length. We are continuing the experiments and will keep updating the content afterwards.
> ### Table 1: Comparing MSSR and PySR on PMLB
> | Algorithm       | $R^2$ | Model Size|
> |-----------------|----------|-------------------|
> | MSSR|0.749727 | 33      |
> | PySR | 0.749435  | 22       |
>
> [1] Cranmer, Miles. "Interpretable machine learning for science with PySR and SymbolicRegression. jl." arXiv preprint arXiv:2305.01582 (2023).

---

> ### Author Response · Authors · 2025-11-20
>
> We are grateful to the reviewer 5Pwr for providing valuable comments and constructive suggestions. This is another part of our response.
>
> **W4.2**: Ablation study limitations: The ablation experiments are limited to a few datasets, which may coincidentally align with the method’s structural assumptions. To demonstrate generalizability, ablations should cover all benchmark categories (PMLB, FSRB, Strogatz).
> **R4.2**: We supplemented the ablation experiments on the PMLB benchmark to ensure coverage of all benchmarks. Additionally, we revised the header of Table 1 and added a benchmark field to clarify it. The results indicate that the combined effect of these components is improved across all benchmarks.
>
>  ### Table 1: Ablation results.
> | Setting | Benchmark | A| B | C|D|E|
> |-----------------|----------|--------------|-------------|-------------|-------------|--------------|
> | Policy Network | | ✔     | ✔ | | ✔     |
> | Sub-expression Library | | ✔     | ✔ | ✔ |      | ✔
> |Evolution|  | ✔     |  | ✔ |      |
> | Dataset| | | | $R^2$ Test
> | feynman_III_15_12 |FSRB | 1.0 |0.7712 |0.9997 |0.2689 |0.2404
> | feynman_I_12_2|FSRB | 1.0 |0.9999| 0.8763 |0.6783 |0.2852
> | feynman_III_19_51|FSRB| 0.9823| 0.8227| 0.8749| 0.2338| 0.4666
> | strogatz_vdp1|Strogatz | 0.9973 |0.9953 |0.8210 |0.3268 |0.2842
> |**596_fri_c2_250_5**| **PMLB** | **0.8498**| **0.7973**| **0.6889** |**0.1683**|**0.6374**
> |Average| |0.9658| 0.8772| 0.8521| 0.3352| 0.3827
>
> **W4.3**: Lack of comparison with GP-ADF and pretrained methods: GP-ADF directly addresses sub-expression reuse and should be included. Additionally, pretrained approaches like SNIP or NeSymReS, which excel in model complexity reduction, are not evaluated.
> **R4.3**:  We have tested GP-ADF[1] on the PMLB benchmark, evaluating both $R^2$ Test and model size. The results are shown in Table 1. Compared with GP-ADF, MSSR achieves higher R² and smaller model sizes. GP-ADF sometimes suffers from an expression length explosion due to excessively nested expressions.
>  ### Table 1: Comparing MSSR and GP-ADF on PMLB
> | Algorithm       | $R^2$ | Model Size|
> |-----------------|----------|-------------------|
> | **MSSR**|**0.749727** | **33**      |
> | GP-ADF| 0.484307 | 172       |
>
> We alse compared MSSR with the reported results of SNIP[2] in the following benchmarks of PMLB. The results is listed in the following table.  Compared to SNIP, MSSR has better average
> $R^2$ and model size.
>
>  ### Table 2: Comparing MSSR and SNIP on PMLB
> | Dataset| **MSSR $R^2$** |**MSSR Model Size **| SNIP $R^2$ | SNIP Model Size|
> |-----------------|----------|-------------------|----------|-------------------|
> | 1096_FacultySalaries | 0.826 |  31      | 0.5607 |39
> | 192_vineyard   | 0.3945  |38            |0.4623|44
> | 225_puma8NH  | 0.6805 | 33          |0.6703|49
> | 228_elusage | 0.7164  | 34     |0.4708|37
> | 547_no2| 0.4803  |33           |0.4680|55
> | 579_fri_c0_250_5| 0.7817 |33             |0.9016|36
> | 591_fri_c1_100_10| 0.867 | 43             |0.7389|38
> | 594_fri_c2_100_5| 0.743 | 37       |0.8039|40
> | 595_fri_c0_1000_10| 0.8189 | 49.5         |0.9111|62
> | 596_fri_c2_250_5| 0.8498 | 55              |0.9035|41
> | 601_fri_c1_250_5| 0.8635 |38              |0.9568|49
> | 609_fri_c0_1000_5| 0.8663 | 45.5               |0.9451|40
> | 611_fri_c3_100_5| 0.8403| 40.5            |0.9609|39
> | 612_fri_c1_1000_5| 0.8794| 45.5	            |0.9206|32
> | 628_fri_c3_1000_5| 0.8916| 40.5          |0.9445|36
> | 656_fri_c1_100_5| 0.8402| 42            |0.9746|47
> | 659_sleuth_ex1714| 0.633| 27            |0.4255|49
> | 665_sleuth_case2002| 0.343| 32.5             |0|41
> | 666_rmftsa_ladata| 0.6784| 30.5           |0.5001|62
> | 687_sleuth_ex1605| 0.415| 30.5            |0.1386|45
> | 712_chscase_geyser1| 0.7497| 41            |0.7092|37
> | Avg | **0.7218** | **38.29**          |0.6841 | 43.71
>
>
> [1] https://github.com/DEAP/deap
> [2] Meidani, Kazem, et al. "Snip: Bridging mathematical symbolic and numeric realms with unified pre-training." arXiv preprint arXiv:2310.02227 (2023).

---

### Official Review · Reviewer_XfqF · 2025-10-29

**Soundness:** 2
**Presentation:** 2
**Contribution:** 2
**Rating:** 6
**Confidence:** 3

**Summary:**

This paper proposes MSSR (Mining Sub-Expression Symbolic Regression) — a novel framework for symbolic regression that aims to reduce the massive combinatorial search space by reusing valuable sub-expressions rather than building expressions from atomic operators.

**Strengths:**

1. Clear and Innovative idea. The paper identifies an under-explored direction — reusing sub-expressions to shrink the symbolic search space. This idea is well justified both theoretically and empirically.

**Weaknesses:**

1. **Unclear modeling justification.**
   The rationale for modeling the problem as a *cooperative Multi-Agent Reinforcement Learning (MARL)* setup is unclear. There appears to be a significant conceptual gap between the motivation of *mining valuable sub-expressions* and the decision to formulate it as a MARL problem. A stronger conceptual or mathematical justification for this modeling choice is needed to make the approach more convincing.

2. **Lack of integration with other frameworks.**
   The proposed idea of reusing sub-expressions should, in principle, be applicable beyond GP—such as within Deep RL, MCTS, or LLM-based symbolic regression frameworks. However, the paper presents results only on a GP-style setup without demonstrating or even discussing how MSSR could be extended or adapted to these other paradigms. Including such discussions or preliminary experiments would substantially strengthen the contribution.

3. **Presentation and formatting issues.**

   * The paper lacks sufficient background on symbolic regression and (multi-agent) reinforcement learning. A concise overview of these topics—along with brief introductions to genetic programming and mutual information—should be added before Section 3.
   * The citation format is inconsistent. All instances of `\cite{}` should be replaced with `\citep{}` to ensure proper inline citation style.
   * Mathematical notation is inconsistent—for example, line 161 uses `\mathcal{R}` while line 163 uses `\mathbf{\mathcal{R}}`. This should be standardized throughout.
   * *Theorem 1* is mislabeled; it is not a formal theorem but rather a derived gradient equation. It should be presented as an equation or proposition with a derivation, **not** as a theorem accompanied by a proof.
   * In Table 5, the variable should be written as `$u_0$` (not `u0`). All mathematical operators should use LaTeX commands such as `$\sin$`, `$\log$`, etc., for proper typesetting consistency.

**Questions:**

Overall, the idea is novel and promising. However, I strongly recommend that the authors carefully proofread the entire manuscript during the rebuttal phase. Without first addressing the numerous writing and formatting issues, additional technical feedback will have limited impact on improving the overall quality of the paper.

---

> ### Author Response · Authors · 2025-11-20
>
> We are grateful to the reviewer XfqF for providing valuable comments and constructive suggestions. Below are our responses to your questions respectively.
>
> **Weaknesses**
> **W1**:**Unclear modeling justification.**
> **W1.1**:  The rationale for modeling the problem as a cooperative Multi-Agent Reinforcement Learning (MARL) setup is unclear. There appears to be a significant conceptual gap between the motivation of mining valuable sub-expressions and the decision to formulate it as a MARL problem. A stronger conceptual or mathematical justification for this modeling choice is needed to make the approach more convincing.
> **R1.1**: We thank the reviewer for the comment. In MSSR, we use three agents to construct each sub-expression: one agent selects the left sub-expression, one selects the operator, and one selects the right sub-expression. These agents cooperate to form a complete sub-expression, and therefore, we use a shared reward to update all three agents. Modeling the problem as a cooperative Multi-Agent Reinforcement Learning (MARL) setup naturally captures this structure: each agent is responsible for a specific component, and the shared reward ensures coordination among agents to generate valid and valuable sub-expressions. We further clarified the model's definition in the manuscript.
>
> The clarified content in the Section 3 of the modified manuscript is listed as follows.
>
> MSSR formalizes the SR problem as a cooperative \textbf{Multi-Agent Reinforcement Learning} (MARL) system $\left \langle  \boldsymbol{\mathcal{N}} , \boldsymbol{\mathcal{S}} ,  \boldsymbol{\mathcal{A}} ,  \boldsymbol{\mathcal{R}}  \right \rangle$.
> The cooperative MARL framework is governed by a joint policy $\boldsymbol{\pi}$, defined as the collection of individual policies:
> $\lbrace \pi_i(a_i|s_i;\theta_i) \mid i \in \{l,o,r\} \rbrace$. $\pi_l$ selects the left sub-expression $S_l$, $\pi_o$ selects the operator $O$, and $\pi_r$ selects the right sub-expression $S_r$, which together compose a sub-expression $\left \langle S_{l},\ O,\ S_{r}\right \rangle$.
>
> **W2**: **Lack of integration with other frameworks**.
> **W2.1**: The proposed idea of reusing sub-expressions should, in principle, be applicable beyond GP—such as within Deep RL, MCTS, or LLM-based symbolic regression frameworks. However, the paper presents results only on a GP-style setup without demonstrating or even discussing how MSSR could be extended or adapted to these other paradigms. Including such discussions or preliminary experiments would substantially strengthen the contribution.
> **R2.1**:  Compared to GP-based methods, Deep RL, MCTS, or LLM-based symbolic regression could potentially generate higher-quality sub-expression libraries. But sampling sub-expressions in these frameworks is very time-consuming. Therefore, we cannot conduct comprehensive evaluations of these algorithms within the limited time available. Nevertheless, we have discussed in the manuscript the possibility of discovering sub-expressions using these paradigms in Section 5. We emphasize the need to balance the time of sampling a sub-expression and its potential value.   This can be viewed as a time–reward trade-off.
>
> The clarified content in the Section 5 of the modified manuscript is listed as follows.
>
> Our future work will focus on integrating various symbolic regression algorithms to mine more meaningful and informative sub-expression libraries. In addition, we will pay particular attention to balancing the efficiency of generating sub-expressions with their overall effectiveness.

---

> ### Author Response · Authors · 2025-11-20
>
> We are grateful to the reviewer XfqF for providing valuable comments and constructive suggestions.This is another part of our response.
>
> **W3**: **Presentation and formatting issues.**
> **W3.1**: The paper lacks sufficient background on symbolic regression and (multi-agent) reinforcement learning. A concise overview of these topics—along with brief introductions to genetic programming and mutual information—should be added before Section 3.
> **R3.1**:  We have expanded Section 2 to provide the necessary background for symbolic regression, (multi-agent) reinforcement learning, genetic programming, and mutual information.
>
> The clarified content in the Section 2 of the modified manuscript is listed as follows.
>
> Symbolic regression (SR) refers to discovering a mathematical expression fitted by
> the given dataset from the vast search space [1,2].
> Genetic Programming (GP) [3,4,5,6,7] is a key method for symbolic regression, encoding expressions and evolving them through operators like crossover, mutation, and selection to achieve better results.
>
> Reinforcement learning (RL) [8] studies how an agent optimizes its decisions to maximize rewards. Cooperative multi-agent reinforcement learning (MARL) allows multiple agents to optimize a shared goal, extending single-agent reinforcement learning [9]. In MSSR, each agent is responsible for decisions in different parts of the sub-expression and uses the same reward for optimization.
>
> Mutual information (MI) [10] and the coefficient of variation (CV) [11] are core metrics used in the MSSR framework to measure correlation and stability. MI is used to measure the dependency between two random variables and remains unchanged under invertible transformations [12] . MSSR uses MI to measure the dependency between sub-expressions and the target, and updates the sub-expression library with those showing higher MI. And, CV is the ratio of standard deviation to the mean [11], which can be used to measure stability. MSSR leverages CV to identify and remove low-value sub-expressions, thus reducing redundancy.
> [1] Schmidt, Michael, and Hod Lipson. "Distilling free-form natural laws from experimental data." science 324.5923 (2009): 81-85.
> [2] Korns, Michael F. "A baseline symbolic regression algorithm." Genetic Programming Theory and Practice X. New York, NY: Springer New York, 2013. 117-137.
> [3] Koza, John R. "Genetic programming as a means for programming computers by natural selection." Statistics and computing 4.2 (1994): 87-112.
> [4] McKay, Robert I., et al. "Grammar-based genetic programming: a survey." Genetic Programming and Evolvable Machines 11.3 (2010): 365-396.
> [5] Ferreira, Candida. "Gene expression programming: a new adaptive algorithm for solving problems." arXiv preprint cs/0102027 (2001).
> [6] Miller, Julian, and Andrew Turner. "Cartesian genetic programming." Proceedings of the Companion Publication of the 2015 Annual Conference on Genetic and Evolutionary Computation. 2015.
> [7] Arnaldo, Ignacio, Krzysztof Krawiec, and Una-May O'Reilly. "Multiple regression genetic programming." Proceedings of the 2014 annual conference on genetic and evolutionary computation. 2014.
> [8] Kaelbling, Leslie Pack, Michael L. Littman, and Andrew W. Moore. "Reinforcement learning: A survey." Journal of artificial intelligence research 4 (1996): 237-285.
> [9]Oroojlooy, Afshin, and Davood Hajinezhad. "A review of cooperative multi-agent deep reinforcement learning." Applied Intelligence 53.11 (2023): 13677-13722.
> [10] Kraskov, Alexander, Harald Stögbauer, and Peter Grassberger. "Estimating mutual information." Physical Review E—Statistical, Nonlinear, and Soft Matter Physics 69.6 (2004): 066138.
> [11] Abdi, Hervé. "Coefficient of variation." Encyclopedia of research design 1.5 (2010): 169-171.
> [12] Zojaji, Zahra, and Mohammad Mehdi Ebadzadeh. "Semantic schema theory for genetic programming." Applied Intelligence 44.1 (2016): 67-87.
>
> **W3.2-3.5**:
> The citation format is inconsistent. All instances of \cite{} should be replaced with \citep{} to ensure proper inline citation style.
> Mathematical notation is inconsistent—for example, line 161 uses $\mathcal{R}$ while line 163 uses $\mathbf{\mathcal{R}}$. This should be standardized throughout.
> Theorem 1 is mislabeled; it is not a formal theorem but rather a derived gradient equation. It should be presented as an equation or proposition with a derivation, not as a theorem accompanied by a proof.
> In Table 5, the variable should be written as $u_0$ (not u0). All mathematical operators should use LaTeX commands such as $\sin$, $\log$, etc., for proper typesetting consistency.
> **R3.2-3.5**: Thanks for your careful comments. We have double-checked the paper to avoid these errors.

---

### Official Review · Reviewer_WFLM · 2025-10-30

**Soundness:** 2
**Presentation:** 2
**Contribution:** 2
**Rating:** 4
**Confidence:** 5

**Summary:**

The paper proposes MSSR, a new symbolic regression algorithm that explicitly identifies and reuses sub-expressions. MSSR trains 3 cooperative agents, tasked with sampling the left sub-expression, operators and right sub-expression in a specific formulation they present in Eq. (1).

**Strengths:**

Novel engineering solution that utilizes a variety of techniques (i.e., information theory, evolutionary computation, reinforcement learning) in an appropriate manner that is competitive.

Paper is organized in an easy-to-read way and is intuitive.

Except for the main results in 4.1.1., Ablation results and PDE discovery case study is a nice addition and is acceptable as-is in my opinion.

**Weaknesses:**

What is the definition of “symbolic recovery rate” defined? There are different definitions in SR literature, it should be stated clearly in the paper. How are the constants treated for equality (i.e., is 0.999 treated effectively the same as 1 in an equation)?

Seem to be missing SR algorithms published post-2021. Can the paper comment on this?

The paper claims that other SR approaches tend to be “overlooking the power of reusing meaningful sub-expressions”. However, it is well-known that SR algorithms, especially evolutionary approaches, reuse meaningful sub-expressions via mechanisms like crossover. Thus, I think the discussion and definition of “overlooking the power of reusing meaningful sub-expressions” needs to be more specific and nuanced to accurately reflect current SR literature.

Contradictory/inconsistent results. Figure 2 and Figure 3 contradict each other. For example, in PMLB, MSSR has the best R2 test, and the model size is smaller than Operon. By the definition of Pareto-optimality, Operon cannot be Pareto optimal, yet it is Pareto optimal in Figure 3. This applies to other algorithms as well. I suspect the issue stems from plotting the average ranks instead, which is not the standard practice for Pareto fronts in general and has been proven to suffer from "rank inversion paradox" for SR benchmarking [1]. An easy fix would be to plot the absolute metrics on the axis instead of the ranks of the metrics.

[1] Fong, K. S., & Motani, M. Pareto-Optimal Fronts for Benchmarking Symbolic Regression Algorithms. In Forty-second International Conference on Machine Learning.

**Questions:**

Please address the weaknesses above. In addition to the weaknesses, below are some questions that could possibly justify a further increase in recommendation score.

Can the paper clarify what happens to the sub-expression library when there is an optimized constant in the sub-expression (e.g., x^2.5)? The examples given in the paper do not have constant in the sub-expression.

What algorithm is used to estimate MI (e.g., KSG [2])? Please discuss and cite the algorithm used along with the settings, because while the definition of MI is not ambiguous, there are many variants of MI estimators, each with potentially very different outputs. And, was the choice of MI estimator justified via experimentation?

[2] Kraskov, A., Stögbauer, H., & Grassberger, P. (2004). Estimating mutual information. Physical Review E, Statistical, Nonlinear, and Soft Matter Physics, 69(6), 066138.

Others:

Instead of MI, would another concept in information theory, unique relevance (UR), be more applicable? This is because 2 sub-expressions with high MI could have high overlap/redundancy.

---

> ### Author Response · Authors · 2025-11-20
>
> We are grateful to the reviewer WFLM for providing valuable comments and constructive suggestions. Below are our responses to your questions respectively.
> **Weaknesses**:
> **W1**:What is the definition of “symbolic recovery rate” defined? There are different definitions in SR literature, it should be stated clearly in the paper. How are the constants treated for equality (i.e., is 0.999 treated effectively the same as 1 in an equation)?
> **R1**:The symbolic recovery rate is the proportion of cases in which the Symbolic Solution is successfully recovered. Following the definition used in SRbench[1], a model $\hat{\phi}(x, \hat{\theta})$ is a Symbolic Solution to a problem with ground-truth model $y = \phi^* \( x,\theta^* \) + \epsilon $ , if $\hat{\phi}$ does not reduce to a constant, and if either of the following conditions is true: 1) $\phi^* - \hat{\phi} = a$; or 2) $\frac{\phi^*}{\hat{\phi}} = b, \quad b \neq 0$, for some constants $a$ and $b$.
> For example, the expression $1.995 x_1 \left(1 - \cos(x_2 x_3)\right)$ is considered a Symbolic Solution to the ground-truth $2 x_1 \left(1 - \cos(x_2 x_3)\right)$, since the two expressions differ only by a non-zero constant scalar. We have also added the corresponding citation to clarify this in our work in Section 3.
> [1] La Cava, William, et al. "Contemporary symbolic regression methods and their relative performance." Advances in neural information processing systems 2021.DB1 (2021): 1.
>
> **W2**:Seem to be missing SR algorithms published post-2021. Can the paper comment on this?
> **R2**: To evaluate accuracy, we have compared our proposed algorithm, MSSR, with 22 baseline algorithms in PMLB, which include uDSR(2022)[1]. However, we have omitted uDSR in the recovery rate experiment. In the modified manuscript, We are currently conducting experiments with uDSR on other benchmarks, which are not yet complete; the current results are shown in Table 1. Compared to uDSR, MSSR has better average $R^2$, model size, and recovery rate. Meanwhile, we have added the PySR (2023) [2] algorithm to the experiments on the PMLB benchmarks, and Figures 2 and 3 have been updated accordingly, which correspond to Table 2.
> ### Table 1: Comparing uDSR and MSSR on FSRB
> | Algorithm       | MSSR $R^2$ | MSSR Model Size| MSSR symbolic recovery rate| uDSR $R^2$ | uDSR Model Size| uDSR symbolic recovery rate
> |-----------------|----------|-------------------|-----------------|----------|-------------------|------------------|
> | feynman_III_12_43 |1 | 5      | 100% |1|20|80%
> | feynman_III_19_51 | 0.9882  |   45    |20%|0.6016|17|0%
> | feynman_II_15_4 | 1  | 9      |80%|1|17|90%
> | feynman_I_11_19 | 1  | 11       |100%|0.7919|19|0%
> | feynman_I_14_3 | 1 | 5      |100%|1|16|80%
> | Avg.| **0.9976**|**15**|**80%**|0.8787|17.8|50%
>
>
> ### Table 2: Comparing MSSR and PySR on PMLB
> | Algorithm       | $R^2$ | Model Size|
> |-----------------|----------|-------------------|
> | MSSR|0.749727 | 33      |
> | PySR | 0.749435  | 22       |
>
> [1] Landajuela, Mikel, et al. "A unified framework for deep symbolic regression." Advances in Neural Information Processing Systems 35 (2022): 33985-33998.
> [2] Cranmer, Miles. "Interpretable machine learning for science with PySR and SymbolicRegression. jl." arXiv preprint arXiv:2305.01582 (2023).
>
> **W3**:The paper claims that other SR approaches tend to be “overlooking the power of reusing meaningful sub-expressions”. However, it is well-known that SR algorithms, especially evolutionary approaches, reuse meaningful sub-expressions via mechanisms like crossover. Thus, I think the discussion and definition of “overlooking the power of reusing meaningful sub-expressions” needs to be more specific and nuanced to accurately reflect current SR literature.
> **R3**: Traditional GP-based evolutionary methods reuse sub-expressions through random selection, crossover, and mutation. However, they cannot explicitly determine which sub-expressions are ''meaningful''. As a result, it is difficult to ensure that sub-expressions from the ground-truth expression are effectively reused. In contrast, MSSR evaluates candidate sub-expressions using mutual information to update the sub-expression library. Sub-expressions that are more relevant to the ground-truth expression are more likely to be selected, thereby achieving targeted reuse of sub-expressions. This design facilitates faster and more reliable discovery of the ground-truth expression.

---

> ### Author Response · Authors · 2025-11-20
>
> We are grateful to the reviewer WFLM for providing valuable comments and constructive suggestions.This is another part of our response.
>
> **W4**: Contradictory/inconsistent results. Figure 2 and Figure 3 contradict each other. For example, in PMLB, MSSR has the best $R^2$  test, and the model size is smaller than Operon. By the definition of Pareto-optimality, Operon cannot be Pareto optimal, yet it is Pareto optimal in Figure 3. This applies to other algorithms as well. I suspect the issue stems from plotting the average ranks instead, which is not the standard practice for Pareto fronts in general and has been proven to suffer from "rank inversion paradox" for SR benchmarking [1]. An easy fix would be to plot the absolute metrics on the axis instead of the ranks of the metrics.
> [1] Fong, K. S., & Motani, M. Pareto-Optimal Fronts for Benchmarking Symbolic Regression Algorithms. In Forty-second International Conference on Machine Learning.
> **R4**: We appreciate the reviewer's comment. Figures 2 and 3 use different metrics: one shows $R^2$  and model size, while the other shows the ranks of $R^2$  and model size. Due to this difference in metrics, the two figures may exhibit slight inconsistencies [1].
> In the revised manuscript, we have added an explanation in Appendix A.6 regarding the use of $R^2$ and model size directly. Table 1 and Table 2 are the corresponding absolute metrics.
> [1] Fong, Kei Sen, and Mehul Motani. "Pareto-Optimal Fronts for Benchmarking Symbolic Regression Algorithms." Forty-second International Conference on Machine Learning.
> ### Table 1: Results for PMLB
> | Algorithm       | $R^2$ | Model Size|
> |-----------------|----------|-------------------|
> | **MSSR**|**0.7497** | **33**         |
> | DSR    | 0.5764  | 5             |
> | gplearn   | 0.6612 | 9          |
> | AFP_FE| 0.6647  | 37       |
> | Operon| 0.7085   |38            |
> | GP-GOMEA| 0.7452 |19             |
> | FEAT| 0.7368 | 57              |
> | KernelRidge| 0.7125 | 110        |
> | XGB| 0.7096| 4577         |
> | EPLEX| 0.7077 | 55              |
> | AdaBoost| 0.6807 | 9680               |
> | RandomForest| 0.6636 | 47967               |
> | MLP| 0.6530| 1402             |
> | AFP| 0.6504 | 26             |
> | uDSR|0.5973   | 18            |
> | ITEA| 0.5893 | 93          |
> | BSR| 0.2979 | 18          |
> | AIFeynman| -1.5279   | 742          |
> | MRGP| 0.4080   | 4020            |
> | FFX|0.3151   | 1098             |
> | LGBM| 0.3121   | 2159            |
> | SBP-GP|0.7285  | 584               |
> | Linear|0.5267 |  5            |
> | GP-ADF|0.4843   | 172         |
> | PySR| 0.7494  | 22           |
> ### Table 2: Results for FSRB and Strogatz
> | Algorithm       | $R^2$ | Model Size|
> |-----------------|----------|-------------------|
> | **MSSR**| **0.9954**| **23**           |
> | AFP    | 0.9839  | 31             |
> | AFP_FE   | 0.9947 | 35          |
> | AIFeynman| 0.0709| 23          |
> | BSR|0.7668  |29          |
> | DSR| 0.9083 |15         |
> | EPLEX| 0.9661 | 44               |
> | FEAT| 0.9569| 119       |
> | FFX| 0.9245| 233        |
> | GP-GOMEA| 0.9976 | 35             |
> | ITEA| 0.9110| 16              |
> | MRGP| 0.5655 |35107               |
> | Operon|  0.9977| 69             |
> | SBP-GP|0.9934  | 607             |
> | gplearn|0.9142| 22           |
>
> **Questions**
> **Q1**:Can the paper clarify what happens to the sub-expression library when there is an optimized constant in the sub-expression (e.g., x^2.5)? The examples given in the paper do not have constant in the sub-expression.
> **R1**: In MSSR, the sub-expression library includes sub-expressions with optimized constants (e.g.,$1.2 \times x_2$). This is necessary for evaluating whether a sub-expression is part of the ground-truth expression using mutual information (MI). To clarify this concept, we have explicitly defined the library $\mathcal{L}$  in the revised manuscript as   $\lbrace x_{1},x_{2},1.2 \times x_{2},sin(x_{3}),\cdots  \rbrace $ and we have updated Figure 1 to illustrate examples of sub-expressions containing constants.

---

> ### Author Response · Authors · 2025-11-20
>
> We are grateful to the reviewer WFLM for providing valuable comments and constructive suggestions.This is another part of our response.
>
> **Q2**:What algorithm is used to estimate MI (e.g., KSG [2])? Please discuss and cite the algorithm used along with the settings, because while the definition of MI is not ambiguous, there are many variants of MI estimators, each with potentially very different outputs. And, was the choice of MI estimator justified via experimentation?
> [2] Kraskov, A., Stögbauer, H., & Grassberger, P. (2004). Estimating mutual information. Physical Review E, Statistical, Nonlinear, and Soft Matter Physics, 69(6), 066138.
> **R2**: MI is used to evaluate the mutual information between the value X of a sub-expression and the target value Y, which defined as $I(X; Y) = \sum_{y \in Y} \sum_{x \in X} p(x, y) \log\left( \frac{p(x, y)}{p(x)p(y)} \right)$. X and Y are continuous random variables, and the mutual information is estimated using a histogram-based method [1].
> [1] Zojaji, Zahra, and Mohammad Mehdi Ebadzadeh. "Semantic schema theory for genetic programming." Applied Intelligence 44.1 (2016): 67-87.
>
> **Q3**: Instead of MI, would another concept in information theory, unique relevance (UR), be more applicable? This is because 2 sub-expressions with high MI could have high overlap/redundancy.
> **R3**: Thank you for your suggestion. This is a good idea. We believe that MI and UR can play different roles and cannot replace each other; however, they complement each other. We use MI to measure the strength of each candidate sub-expression’s relationship to the target, which helps determine whether it could be part of the ground-truth expression. UR can be used as a complementary measure to quantify redundancy between sub-expressions, ensuring that the sub-expression library remains unique and reducing redundancy.

---

### Author Response · Authors · 2025-11-20

We thank all reviewers for their valuable comments. Due to the time constraints of this round, our current response includes only part of the extended experimental results. Based on these available results, we have addressed the questions related to the extended experiments, and we have incorporated the corresponding partial results into the revised manuscript. Before the deadline for submitting the final manuscript (December 3, 2025), we will include the complete extended experimental results in the final revised version of the paper.

---

### Author Response · Authors · 2025-12-03

Dear reviewers, we have completed experiments using the uDSR algorithm on the FSRB and Strogatz benchmarks and have computed the symbolic recovery rate. The latest results have been incorporated into the manuscript and are shown in Figures 2 and 3, as well as Figure 6 in the appendix. The results indicate that MSSR consistently ranks among the top algorithms in balancing model size and accuracy on the FSRB and Strogatz benchmarks, and it also demonstrates a competitive symbolic recovery rate. We hope that these additional experiments address the previous concerns. All other questions have been answered in our earlier responses. We sincerely thank the reviewers for their valuable comments.

---

### Meta-Review · Area_Chair_pSo1 · 2026-01-02

**Summary:**

Key criticisms were
* not embedding the work enough in existing literature
* unclear expressions such as “symbolic recovery rate”
* unclear explainations of why this model was chosen
* insufficient numerical experiments

**Reviewer Concerns:**

I believe that the authors did address the concerns of the reviewers to a certain extent. But, for instance, the numerical experiments, in particular the ablation experiments, are still not comprehensive enough.

**Reviewer Scores:**

It’s really hard to say how any reviewer would have changed their score if they had taken part more fully in the discussion. Without hearing it from them directly, anything we write here would just be guesswork.

For reference, the scores were 2, 4, and 6. And even in a best-case scenario where everyone bumped their score by +1, the overall decision would still have been a reject.

---

### Decision · Program_Chairs · 2026-01-26

Reject